# Contamination by Trace Elements and Oxidative Stress in the Skeletal Muscle of *Scyliorhinus canicula* from the Central Tyrrhenian Sea

**DOI:** 10.3390/antiox12020524

**Published:** 2023-02-19

**Authors:** Mariacristina Filice, Francesca Romana Reinero, Maria Carmela Cerra, Caterina Faggio, Francesco Luigi Leonetti, Primo Micarelli, Gianni Giglio, Emilio Sperone, Donatella Barca, Sandra Imbrogno

**Affiliations:** 1Department of Biology, Ecology and Earth Sciences, University of Calabria, 87036 Rende, Italy; 2Sharks Studies Center—Scientific Institute, 58024 Massa Marittima, Italy; 3Department of Chemical, Biological, Pharmaceutical and Environmental Sciences, University of Messina, 98166 Messina, Italy

**Keywords:** sharks, trace elements, antioxidant enzyme, lipid peroxidation, protein carbonyl, leukocytes

## Abstract

Marine pollution, due to the regular discharge of contaminants by various anthropogenic sources, is a growing problem that imposes detrimental influences on natural species. Sharks, because of a diet based on smaller polluted animals, are exposed to the risk of water contamination and the subsequent bioaccumulation and biomagnification. Trace elements are very diffuse water pollutants and able to induce oxidative stress in a variety of marine organisms. However, to date, studies on sharks are rather scarce and often limited to mercury. In this context, the present study aimed to analyze the accumulation of trace elements and their putative correlation with the onset of an oxidative status in the muscle of the lesser spotted dogfish *Scyliorhinus canicula*, from the Central Mediterranean Sea. Ecotoxicological analysis detected the presence of Pb, As, Cd, Mn, Zn, Ni, Cu, and Fe; no significant differences were observed between sexes, while a negative correlation was found between Pb and animal length. Analysis of oxidative stress markers showed either positive or negative correlation with respect to the presence of trace elements. Lipid peroxidation (TBARS) positively correlated with Zn, Ni, and Fe; SOD enzyme activity negatively correlated with Cu and Ni; LDH was negatively correlated with Fe and positively correlated with Pb. Moreover, positive correlations between the leukocyte count and Mn and Zn, as well as with LDH activity, were also observed. The data suggested that, in sharks, trace elements accumulation may affect oxidant and antioxidant processes with important outcomes for their physiology and health.

## 1. Introduction

Aquatic environment contamination by trace elements, resulting from geological and biological cycles, as well as from anthropogenic activities, represents a severe and growing challenge for resident organisms [1,2,3,4]. Both non-essential trace elements (e.g., arsenic (As), mercury (Hg), lead (Pb), and cadmium (Cd)), without known physiological functions, and essential trace elements (e.g., zinc (Zn), copper (Cu), nickel (Ni), and iron (Fe)), necessary for basic physiological processes, are able to accumulate in fish tissues [1,5,6,7,8], with potential detrimental effects on the physiology of target organs. Trace elements can interact with diatomic molecular oxygen from metabolic pathways to produce reactive oxygen species (ROS), such as the superoxide anion radical (O_2_^•−^), the hydrogen peroxide (H_2_O_2_), the hydroxyl radical (^•^OH), and the superoxide radical (O_2_). When ROS production exceeds removal, antioxidant defenses can be deactivated; this results in oxidative stress and macromolecule damage (lipid peroxidation, protein degradation, and disruption of DNA), cellular dysfunction, and/or death [9]. A correlation between trace elements accumulation and oxidative stress has been reported in aquatic organisms [1,10,11,12,13,14,15,16]. However, several species show the ability to prevent oxidative damage caused by trace elements. This may occur by scavenging and/or neutralization (e.g., through metallothionein), by increasing the activity of antioxidant enzymes (i.e., superoxide dismutase (SOD), catalase (CAT), and glutathione S-transferase (GST)), and by excretion [1,13].

Sharks, due to the high position in the food chain, slow growth rates, and low elimination abilities [10,17], are able to bioaccumulate, or worse, to biomagnify several elements through the food chain [5,15,18,19,20]. So far, information regarding the influence of the exposure to trace elements on shark health is scarce and fragmentary, often obtained in a limited number of species or in a restricted region of the world [21,22]. Preliminary data revealed that in several species, such as the nurse shark and the spiny dogfish shark, trace elements, such as Pb, As, and Cd, affect reproduction, cardiac performance, homeostasis balance, and vascular dynamics [23,24,25]. Little is known on the putative impact of trace elements accumulation on the oxidative status of animals. In this regard, the few data available, mainly derived from studies performed on epipelagic and demersal species (e.g., blue sharks [1], mako sharks [12], the Atlantic sharpnose shark [10]), suggested a correlation between several trace elements and the activity of antioxidant enzymes (e.g., SOD, CAT, and glutathione peroxidase (GPx)), as well as the levels of oxidation products (e.g., protein carbonylation and lipid peroxidation). However, a significant variability, possibly depending on environmental exposure and/or behavioral differences between populations, sex, or maturity stage, has emerged.

The lesser spotted dogfish *Scyliorhinus canicula* (Linnaeus, 1758) is a small benthic shark, widely distributed in the Mediterranean Sea, in which it represents one of the most abundant species, and in the North-Eastern Atlantic Ocean. It can be found from the intertidal zone to the continental slopes, from 10 to 780 m, with highest abundances at depths of 200–500 m [26,27]. It is an opportunistic and a generalist scavenger mesopredator that feeds mostly on small invertebrates, such as crustaceans and cephalopods, and on small fishes [28]. *S. canicula* is a robust species that well tolerates stress conditions and parasitic load [5]. For this reason, it represents a model useful to assess the phenotypic resilience to both environmental and anthropogenic stressors. To date, the risk assessment has been mainly based on population reduction due to fishing overexploitation [29,30], with limited attention on the effects of environmental challenges due to the increasing pollution of the Mediterranean Sea. Recently, Reinero and coworkers [5] detected the presence of trace elements in vertebrae, skin, and liver of *S. canicula* from the Central Tyrrhenian Sea. However, very few information is available on the skeletal muscle that, for its high activity during continuous swimming, is susceptible of bioaccumulation, with potential crucial outcomes for whole animal fitness [31]. Several shark species, such as *S. canicula* in the Mediterranean, are edible. Thus, contaminated skeletal muscle may represent a major route for transfer of trace elements into humans [32], with possible aftermath for population health.

Based on these observations and by integrating mass spectrometer and biomolecular evaluations, this study aimed: (i) to determine the concentrations of trace elements (Pb, As, Mn, Zn, Ni, Cu, Cd, and Fe); (ii) to quantify main indicators of oxidative stress; and (iii) to assess the potential correlation between trace elements and oxidative stress indicators in the skeletal muscle of the lesser spotted dogfish *S. canicula* from the Central Tyrrhenian Sea. 

## 2. Materials and Methods

### 2.1. Sampling Collection

Sampling was carried out in January 2018 at 12 nautical miles off Rocchette Punta Ala (Tuscany, Italy), between the islands of the Tuscan Archipelago (Elba Island, Giglio Island, Pianosa Island, Montecristo Island, and Capraia Island) and the Gulf of Follonica (geographic coordinates: 42.565655° N, 10.601167° E) (Figure 1), in collaboration with the local fishing boat “Mare Blu”. The chondrichthyan specimens enrolled in the present work were obtained from commercial fisheries. The activity was conducted with the observation of the Regulation of the European Parliament and the Council for fishing in the General Fisheries Commission for the Mediterranean (GFCM) Agreement area and amending Council Regulation (EC) No. 1967/2006. All procedures were carried out with the approval by the “Ministero della transizione ecologica—MiTE” (n. authorization 0008263, 25/01/2022).

In total, 24 specimens of *S. canicula* (*n* = 18 females and *n* = 6 males) were collected by commercial bottom trawling at a 150 m depth and transferred to a laboratory for subsequent processing. Total length and weight were measured for each sample, while sex was established by the presence of claspers in males and their absence in females. Maturity was confirmed by the presence of oocytes and eggs in females and calcified claspers and sperm in males [33]. For the leukocyte count, a blood sample was collected from the caudal vein of each specimen and immediately placed at +4 °C [5]. Specimens were then dissected, and samples of muscle, taken from the area near the dorsal fin, were removed from each sample and stored at −20 °C for ecotoxicological and oxidative stress analyses.

### 2.2. Hematological Analyses

Blood samples were analyzed using the DasitSysmex SX-1000i instrument; the leukocyte formula was obtained by fluorescence flow cytometry [34].

### 2.3. Trace Elements

Analysis of trace elements was performed by using a combination of acid attacks and an Elan DRCe (Perkin Elmer/SCIEX) Inductively Coupled Plasma Mass Spectrometer (ICP-MS), which is routinely used to determine almost all chemical elements in solid or soft tissues [5,20,35].

In total, 24 muscle samples were removed from 24 different sharks. Samples were first dehydrated under a hood for 24 h at a laminar flow without pulverization. They were then weighed in Teflon containers on a high-precision analytical scale after its calibration. Subsequently, samples were placed in 10 mL of an ultra-pure nitric acid (HNO_3_) solution, digested in a CEM MARS-5 microwave oven (50 min at 180 °C; power: 400 W; pressure: 300 PSI) and evaporated on a pre-heated plate at 200 °C. They were brought to a volume with ultrapure water in flasks of known volume (50 or 100 mL depending on the weight of the sample) and stored at −4 °C. The same procedure was applied to prepare a Tort (lobster hepatopancreas)-certified reference material (CRM), used as an unknown sample for quality control standards during the analytical sequence. After acid attack, samples were analyzed with the ICP-MS by introducing them into the spectrometer through a peristaltic pump and transformed into an aerosol by a nebulizer supplied with the instrument. In total, 46 trace elements were analyzed, but only Pb, As, Mn, Zn, Ni, Cd, Cu, and Fe were taken into consideration, because of the very low concentration of all the other elements.

### 2.4. Oxidative Stress Indicators

Evaluation of oxidative stress biomarkers was performed as described in detail by Filice et al. [36]. Muscle samples (N = 24) were homogenized in cold Tris/HCl buffer (100 mM; pH 7.2), containing a mixture of protease inhibitors. An aliquot of homogenates was used to determine TBARS levels; the remaining part was centrifuged at 5000× *g* for 5 min at 4 °C; the supernatant was tested for protein oxidation. Protein concentration was determined with the Bradford method by using a commercial kit (Bio-Rad Laboratories) and bovine serum albumin (BSA) as a standard.

#### 2.4.1. Lipid Peroxidation

To assess lipid peroxidation (LPO), the concentration of 2-thiobarbituric acid-reacting substances (TBARS) was measured according to Tkachenko and Grudniewska [37]. Briefly, a reaction mixture containing the sample homogenate (0.2 mL, 10% *w*/*v*), 2-thiobarbituric acid (TBA; 0.8%, 0.2 mL), and trichloroacetic acid (TCA; 20%, 0.2 mL) was boiled in a water bath at 100 °C for 10 min and then centrifuged at 7000 rpm for 10 min. The supernatant was assessed at 540 nm to determine TBARS levels by measuring malondialdehyde (MDA) content, the major lipid oxidation product. TBARS values were expressed as MDA concentration (μM) per gram of tissue (MDA extinction coefficient: 156,000 M^−1^ cm^−1^).

#### 2.4.2. Protein Oxidation

The levels of oxidatively modified protein (OMP) were evaluated by measuring carbonyl groups content with the traditional 2,4-dinitrophenylhydrazine (DNPH) method [38]. Aliquots of the supernatant were incubated at room temperature for 1 h with 10 mM DNPH in 2 M HCl and then precipitated with 2 volumes of TCA. The solution was centrifuged for 20 min at 7000 rpm; to remove DNPH excess, the pellet was washed thrice with ethanol-ethyl acetate (1:1; *v*/*v*); then, it was dissolved in 6M guanidine in 2N HCl. The concentration of carbonyl groups was measured spectrophotometrically at 370 nm (aldehydic derivates) and at 430 nm (ketonic derivates) using an extinction coefficient of 22,000 M^−1^ cm^−1^. Results were expressed as nmol per mg protein.

#### 2.4.3. Superoxide Dismutase (SOD) Activity

To determine SOD activity, the pyrogallol method of Marklund and Marklund [39], modified by Tresnakova et al. 2023 [40], was used. In brief, the SOD-dependent inhibition of the auto-oxidation of pyrogallol at pH 8.20 was assayed spectrophotometrically at 420 nm and 25 °C. The reaction was prepared in 50 mM Tris-HCl, 1 mM EDTA, and 0.2 mM pyrogallol and monitored every 30 s for 5 min. One unit of SOD activity was defined as the amount of the enzyme that inhibits 50% of pyrogallol auto-oxidation. Results were expressed in U/mL protein.

### 2.5. Lactate Dehydrogenase Assay

Muscle samples were homogenized in a cold homogenization buffer (0.025 M EDTA, 0.01 M NaCl, 0.01 M Tris-HCl (pH 7.4), and 0.1% Triton) containing a protease inhibitors mixture and then centrifuged at 10,000× *g* for 10 min at 4 °C. LDH activity was determined as described by Imbrogno et al. [41], by monitoring NADH oxidation at 340 nm in a 3 mL reaction mixture containing potassium phosphate (0.1 M, pH 7.5), sodium pyruvate (0.66 mM), and NADH (0.2 mM), supplemented with 5 μL of the tissue supernatant. All assays were performed at 25 °C. LDH activity was expressed as UI/mL protein.

### 2.6. Data Analysis

The statistical significance of correlation analyses was assessed using linear correlation tests: not significant (N.S., *p* > 0.05), significant (*: 0.01 ≤ *p* ≤ 0.001), very significant (**: 0.001 ≤ *p* < 0.0001), extremely significant (***: *p* ≤ 0.0001). Statistical tests were carried out in Instat 3.0 for Mac.

## 3. Results

A total of 24 *S. canicula* (*n* = 18 females; *n* = 6 males) were sampled in January 2018 in the Central Tyrrhenian Sea. All specimens were found to be sexually mature. Biometric evaluations revealed an average total length (TL) (±sd) of 39.77 ± 2.95 cm (range: 33–47 cm) and a total weight (TW) (±sd) of 200.06 ± 51.56 g (range: 135–317.50 g) (Appendix A). No significant differences were detected between the two sexes, although a tendency was observed with males being larger and heavier than females.

### 3.1. Trace Elements Concentrations

Trace element concentrations measured in muscle samples of *S. canicula* are summarized in Table 1. No differences were observed between males and females. Only Pb levels were found to be significantly lower in animals with higher TL (*p* < 0.001).

### 3.2. Oxidative Status

The average values for TBARS levels, protein carbonyl concentrations, and SOD and LDH activity, measured in muscle homogenates of *S. canicula* are reported in Table 2.

Statistical correlation analysis revealed that protein carbonyl levels (OMP-ALD and OMP-KET) were not influenced by sex but were significantly lower in specimens with higher TL (*p* < 0.01 in both cases). TBARS levels and LDH activity were higher in females (*p* < 0.001 for TBARS and *p* < 0.01 for LDH), while SOD levels were higher in males (*p* < 0.01); no correlation was observed between these markers and TL (Table 3).

Significant correlations were found between oxidative stress indicators (Table 4): TBARS were negatively correlated with SOD (*p* < 0.01); OMPALD and OMPKET were positively correlated with each other (*p* < 0.0001) and with SOD (*p* < 0.01); LDH was not correlated with any of the other oxidative stress indicators.

### 3.3. Leukocyte Count

Blood samples to perform the leukocyte count (10^3^/µL) was collected from 16 sharks. The average value (±sd) of white blood cells (WBCs) was 186.35 ± 51.41 10^3^/µL (range: 77.56–274.54 10^3^/µL).

### 3.4. Correlations between Trace Elements, Oxidative Stress Indicators, and the Leukocyte Count

The correlations between trace elements and oxidative stress indicators are shown in Table 5. TBARS positively correlated with Zn, Ni, and Fe (*p* < 0.01 in all cases), while SOD activity was negatively correlated with Cu and Ni (*p* < 0.01 in both cases). LDH was negatively correlated with Fe (*p* < 0.01) and positively correlated with Pb (*p* < 0.01).

The leukocyte count showed a significant and positive correlation (Table 6) with Mn and Zn (*p* < 0.01), as well as with LDH activity (*p* < 0.01).

## 4. Discussion

The present study highlighted the correlations between trace elements accumulation and markers of oxidative status in the muscle of the lesser spotted dogfish *S. canicula*, collected at a 150 m depth in the Central Tyrrhenian Sea.

As revealed by animal inspection, all specimens were sexually mature, and a prevalence of females with respect to males was observed in the whole sample. This different sex ratio is not unusual in sharks, being possibly related to the breeding behavior of the species. For example, unisexual aggregations of *S. canicula* have been observed in certain geographical areas and at different times of the year [17,29,42,43]. The biometric analysis of the samples used in this study showed no significant differences between males and females in terms of length and weight. However, a tendency was observed, with males longer and heavier than females. This is interesting since, despite females are usually larger than males, sometimes the opposite is observed due to a differential growth rate between the sexes [42]. In fact, after sexual maturity, females spend more energy into egg development and ovulation and this results in a slower growth, while males of the same size maintain a constant growth rate [42]. By spectrometer analysis, the presence of trace elements was evaluated in the skeletal muscle of all sharks here considered. The comparison of the data with those obtained on different anatomical districts of the same animals and already reported [5] revealed that the concentrations of trace elements were similar to those found in the vertebrae but higher than those detected in the skin and liver, suggesting that, in the lesser spotted dogfish, the vertebrae and the skeletal muscle may represent a major bioaccumulating district of the body. Exceptions are represented by Ni and Fe that, due to their high affinity with placoid scales [44,45,46], are mainly accumulated in the skin [5] and by Cu that, in association with Zn and Cd, is mainly accumulated in liver [47].

To the best of our knowledge, this is the first description of trace elements accumulation in the muscle of *S. canicula* from individuals collected in the wild in the Mediterranean. In fact, most of the ecotoxicological analyses carried out in *S. canicula* in target organs and tissues, including the muscle, are from studies performed during artificial experiments in a controlled environment [45,48,49]. This prevents a conclusive comparison of our data with already available observations. At the same time, our results may be considered a reference point for future investigations with the ICP-MS technique on small benthic or demersal mesopredator sharks coming from other Mediterranean areas and naturally exposed to aquatic contaminants. With the exception of Cu, our results showed higher concentrations of trace elements in the muscle of *S. canicula* than in the same tissue of *Galeus melastomus* (Rafinesque, 1810) from the Northern Tyrrhenian Sea [50], *G. melastomus* and *Centrophorus granulosus* (Bloch and Schneider, 1801) coming from the Eastern Mediterranean Sea [51], and *Mustelus mustelus* (Linnaeus, 1758) coming from the southeastern waters of the Mediterranean Sea [52]. With respect to the data reported for other Mediterranean sharks, the relatively higher levels of trace elements we detected in the skeletal muscle of *S. canicula* could be in part related to its sedentary lifestyle, with the consequent chronic exposure to bioaccumulation when inhabiting degraded environments [5]. This possibility is supported by data in benthic or demersal sharks showing a high vulnerability to pollution than pelagic and migratory animals [30]. Of note, the high concentrations of trace elements we discovered are consistent with the data from ARPAT [53] that reported high levels of pollution in waters around the Tuscan Archipelago. As already shown for other target tissues of the same species, such as vertebrae, skin, and liver [5], for the skeletal muscle, no significant correlation was observed between contamination levels and animal sexes. Only Pb was negatively correlated with the size of the sharks, being lower in animals with higher TL. This trend is well-known in fishes [54,55]. In *S. canicula*, this may be sustained by a growth-related change of the diet [17] and the higher detoxifying activity of the rectal gland observed in relation to animal growth (and the consequent increase in TL) that may contribute to excrete some toxic trace elements, as in the case of Pb [45,46]. Of note, in the skeletal muscle, the levels of Mn and Zn significantly and negatively correlated with the leukocyte count. A decrease in leukocytes has been demonstrated in the vertebrae of *S. canicula* in relation to an increased concentration of As [5]. In different marine and terrestrial animals (e.g., glaucous gull *Larus hyperboreus* [56]; African common toad *Amietophrynus regularis* [57]; green frog *Pelophylax synkl. hispanicus* [35]), the increase of the parasitic load, together with the accumulation of trace elements, has been linked to a general condition of immunosuppression. Based on our results, it can be hypothesized that in *S. canicula*, the presence of Mn and Zn could compromise the activity of the immune system.

The exposure of marine organisms to pollutants results in molecular alterations and in structural and functional changes that can be detrimental for animal fitness [40,58,59]. The evaluation of these events takes advantage of the analysis of well-established biological markers. Among others, the activity of antioxidant enzymes and the levels of oxidation products (TBARS and protein carbonyls) are very useful to detect alterations of the oxidative status of the whole organism and of specific tissue targets [12]. Antioxidant enzymes prevent oxidative tissue damage by minimizing either the production of ROS or their interaction with other molecules [60]. In the present study, we observed in the skeletal muscle of *S. canicula* an activity of SOD which is in line with values reported in the same tissue for other pelagic shark species, such as the blue sharks *Prionace glauca* [11] and the shortfin mako *Isurus oxyrinchus* [12]. This is the first study reporting reference values for SOD activity in *S. canicula*. Only one work analyzed antioxidant efficiency in the lesser spotted dogfish, with reference values relative only to CAT and GST [61]. Different from the blue shark, in *S. canicula* SOD activity is sex-correlated, with higher values in males than females. Moreover, SOD is inversely correlated to lipid peroxidation (TBARS), whose levels in males are lower than in females. It cannot be excluded that the differences in the levels of oxidative stress indicators in target tissues are related to the reproductive status and to sex, since during gestation females have higher metabolic rates, which may increase ROS production by affecting the balance between pro-oxidants and antioxidants factors [60,62]. The levels of TBARS here reported are in line with that previously described for *S. canicula* [63,64]. Of note, in the skeletal muscle of cartilaginous fishes, including *S. canicula*, TBARS levels are significantly higher than those reported in teleost [63]. This has been attributed to the presence in elasmobranchs of squalene, a low-density polyunsaturated hydrocarbon necessary to ensure neutral buoyancy [65]. Notably, in *S. canicula*, high lipid peroxidation levels are paralleled by lower antioxidant activity [64]; this may support the negative correlation between TBARS and SOD observed in the present study.

When the oxidative status was correlated with the presence of Ni, Zn, and Fe, we found a positive correlation with TBARS levels. In aquatic animals, exposure to these metals, particularly Fe and Ni, induces oxidative stress by inhibiting the activity of antioxidant enzymes [66,67] and by promoting ROS generation. This, in turn, may increase lipid peroxidation [68,69]. For example, in *Oryzias latipes* embryo, exposure to nano-iron inhibits antioxidant enzymes activity in a dose-dependent manner with consequent induction of lipid peroxidation [66]. Iron overload increases lipid peroxidation also in liver and heart tissues of *Clarius gariepinus* [70], as well as in gills and liver of *Labeo rohita* [69]. Similarly, the capacity of Ni to induce lipid peroxidation has been demonstrated in various fish species, such as *Salvelinus namaycush* [71], *Coregonus clupeaformis* [72], *Cirrhinus mrigala* [67], *Carassius auratus* [73], and *Prochilodus lineatus* [68]. Of note, in all these species, the Ni-dependent increase of TBARS levels was associated with a significant reduction in antioxidant enzyme activity, including SOD, a trend similar to that observed in the present study. This suggests that in shark skeletal muscle Ni is able to inhibit the antioxidant capacity of SOD and to induce lipid peroxidation, although it is present at lower concentration than in other tissues [5]. No evidence showing a direct correlation between Ni and Fe and oxidative markers is available in sharks. However, in the muscle of mako sharks from Baja California Sur, the activity of antioxidant enzymes, such as SOD and CAT, is significantly correlated with As, and other correlations were found between GPx and Pb, As, and Cd [12]. Similarly, in the liver of blue sharks from Baja California Sur, a significant correlation was found between CAT, TBARS, As, and Cd and between SOD, Zn, and Pb in the kidneys [1].

Together with lipid peroxidation, carbonyl derivatives have become one of the most frequently used markers of oxidative stress [74]. An increase of protein carbonyl (PC) levels has been observed in some marine species exposed to pollutants [75,76]; however, only few studies assessed the potential of this biomarker in shark species [11,12]. Higher PCs amounts were found in muscle samples from males of *P. glauca*, when compared to females [11], while a different tissue expression was observed in juvenile individuals of *I. oxyrinchus*, with the highest PCs levels detected in the kidneys, followed by liver and muscle [12]. In our study, the levels of oxidatively modified proteins, estimated by OMP-ALD and OMP-KET, resulted independent of sex but inversely correlated to TL. The nature of this correlation remains unclear. However, as proposed for trace elements, the possibility that the antioxidant capacity to remove oxidative products over time increases with animal size cannot be excluded. The positive correlation that we observed between OMP and SOD activity may support this hypothesis. Moreover, no correlation with OMP levels and trace elements has been found in this study. Although different studies analyzed this biomarker in response to pollutants, a correlation between PC and trace elements was reported only for *P. glauca* [1], in which PC levels are negatively correlated with Fe and positively correlated with As and Cu. In this context, the use of PCs as biomarkers in studies aimed to assess the effects of pollution in elasmobranchs requires to be further confirmed.

Lactate dehydrogenase (LDH), the terminal enzyme in vertebrate’s anaerobic glycolysis, has recently emerged as an important biomarker useful to determine, in fish, tissue and cellular damage caused by pollutants [77]. Trace metals exposure on LDH activity in fish is controversial. An increased LDH activity in response to different trace metals (Cu, Cd, Pb, and Hg) has been reported in *Cyprinus carpio* [78], *P. lineatus* [79], *Synechogobius hasta* [80], and *Catla catla* [81]. Conversely, a reduction of enzyme activity has been observed in *Sparus aurata* and *Clarias gariepinus* exposed to Cu and Pb, respectively [82,83,84]. In our study, LDH was correlated positively with Pb and negatively with Fe. In aquatic species, exposure to Pb is proposed to contribute to a reduced oxygen uptake rate [85,86] and a decreased activity of glycolytic enzymes [87,88]. Particularly in tissue with high metabolic rates, as brain and muscle, this may induce the activation of anaerobic pathways. As reported above, the skeletal muscle of *S. canicula* accumulates Pb more than other tissues [5]. The positive correlation we observed between this metal and LDH activity may suggest that in the skeletal muscle, anaerobic pathways are induced for energy release. This mechanism was already observed in fish in response to mercury exposure [89,90,91]. Different from Pb, information on the effect of Fe accumulation on LDH activity in fish is currently very scarce. Only one study carried out in trout showed an increase in LDH activity in the serum of tilapia *Oreochromis niloticus* exposed to iron oxide nanoparticles [92]. Whether in *S. canicula* Fe accumulation may negatively affect muscle energy metabolism deserves further investigation.

Our study revealed a positive correlation between muscle LDH activity and serum leukocytes. LDH not only functions as a marker of energy metabolism but, together with other oxidative and metabolic enzymes, it also represents an indicator of stress [93] that may reflect animal health. In addition, blood cells count is indicative of the health status of an animal under challenging conditions, including contaminants exposure [94]. It is known that trace elements bioaccumulation is associated with immunosuppression in different shark species, including *S. canicula* [5], thus making animals more sensitive to parasite infestation. Our data pave the way to further investigations aimed to assess the value of the correlation between muscle LDH activity and serum leukocytes as a marker of health in fish under metals bioaccumulation.

## 5. Conclusions

In this work, we found that the skeletal muscle of a Mediterranean population of *S. canicula* contained eight trace elements (Pb, As, Cd, Mn, Zn, Ni, Cu, and Fe), whose amounts, except for Ni, Cu, and Fe, were higher than those present in skin and liver. The correlation between the levels of these contaminants, the leukocytes count, and biological markers of oxidative stress suggested a possible role of Pb, Zn, Ni, and Fe as predictive markers of oxidative stress in the lesser spotted shark. In addition, Mn and Zn, together with LDH, could be considered indicators of animal health.

Marine pollution, mostly resulting from urban and industrial activity, is going to be a serious growing problem. Although, at the moment, the risk assessment of *S. canicula* mainly relies on fishing overexploitation, the results of the present study suggest that the impact of environmental pollution must be taken into serious consideration. In this regard, analysis of the correlation between the levels of trace elements and the stress-related response could represent a useful tool for monitoring the health status of this species in the wild.

## Figures and Tables

**Figure 1 antioxidants-12-00524-f001:**
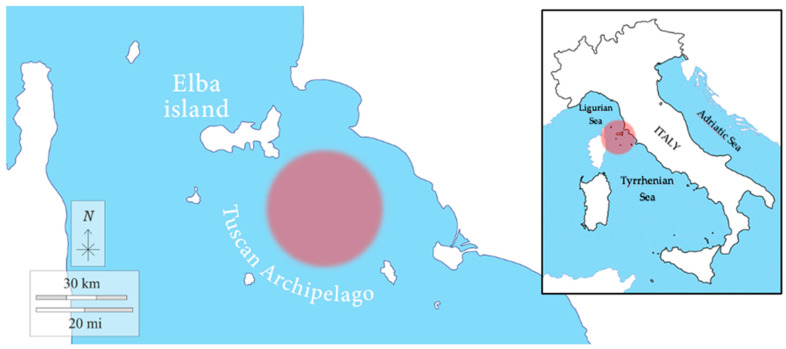
Sampling area. The fishing area of the samples is indicated by the red circle in the figure.

**Table 1 antioxidants-12-00524-t001:** Trace elements concentrations (mg Kg^−1^) in muscle samples of the lesser spotted dogfish. Data are shown as the average value, the standard deviation (sd), and the range (minimum–maximum).

	Pb	As	Mn	Zn	Ni	Cu	Cd	Fe
Mean ± SD (Min–Max)	5.16 ± 11.7 (0–48.23)	265.23 ± 328.7 (24.57–1459.4)	141.45 ± 96 (26–382)	100.23 ± 68.72 (0–321.89)	6.59 ± 16.54 (0–81.23)	8.59 ± 13.25 (0.17–64.75)	0.17 ± 0.52 (0–2.44)	100.40 ± 287.44 (0–1444.3)

**Table 2 antioxidants-12-00524-t002:** Oxidative stress indicators in the muscle tissue of the lesser spotted dogfish *S. canicula*. Data are shown as the average value, the standard deviation (sd), and the range (minimum–maximum).

	TBARS (nmol/g Tissue)	OMP-ALD (nmol/mg Prot)	OMP-KET (nmol/mg Prot)	SOD (U/mL Prot)	LDH (Ul/mL Prot)
Mean ± sd (Min–Max)	77.86 ± 23.90 (39.59–126.12)	4.56 ± 1.10 (1.78–6.51)	1.46 ± 0.50 (0.44–2.30)	33.82 ± 6.96 (9.77–40)	52.53 ± 16.50 (23.33–88.14)

**Table 3 antioxidants-12-00524-t003:** Differences in oxidative stress indicators in the muscle of the lesser spotted dogfish *S. canicula* between sexes and correlation with TL. N.S.—not significant; * *p* < 0.01; ** *p* < 0.001.

	SEX	TL
TBARS	F > M **	N.S.
OMP-ALD	N.S.	*
OMP-KET	N.S.	*
SOD	M > F *	N.S.
LDH	F > M *	N.S.

**Table 4 antioxidants-12-00524-t004:** Correlations between oxidative stress indicators in the muscle of the lesser spotted dogfish *S. canicula*. N.S.—not significant; * *p* < 0.01; *** *p* < 0.0001.

	TBARS	OMP-ALD	OMP-KET	SOD	LDH
TBARS	/	N.S.	N.S.	*	N.S.
OMP-ALD		/	***	*	N.S.
OMP-KET			/	*	N.S.
SOD				/	N.S.
LDH					/

**Table 5 antioxidants-12-00524-t005:** Correlations between oxidative stress indicators and trace elements accumulation in the muscle of the lesser spotted dogfish *S.canicula*. (N.S.—not significant; * *p* < 0.01).

	Pb	As	Mn	Zn	Ni	Cu	Cd	Fe
TBARS	N.S.	N.S.	N.S.	*	*	N.S.	N.S.	*
OMP-ALD	N.S.	N.S.	N.S.	N.S.	N.S.	N.S.	N.S.	N.S.
OMP-KET	N.S.	N.S.	N.S.	N.S.	N.S.	N.S.	N.S.	N.S.
SOD	N.S.	N.S.	N.S.	N.S.	*	*	N.S.	N.S.
LDH	*	N.S.	N.S.	N.S.	N.S.	N.S.	N.S.	*

**Table 6 antioxidants-12-00524-t006:** Correlations between oxidative stress indicators and trace elements accumulation with the leukocytes count (N.S.—not significant; * *p* < 0.01).

Parameter	*p*
Pb	N.S.
As	N.S.
Mn	*
Zn	*
Ni	N.S.
Cu	N.S.
Cd	N.S.
Fe	N.S.
TBARS	N.S.
OMP-ALD	N.S.
OMP-KET	N.S.
SOD	N.S.
LDH	*

## Data Availability

Data are available on request due to restrictions, e.g., privacy or ethical.

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
