# Peer review of "Contamination by Trace Elements and Oxidative Stress in the Skeletal Muscle of Scyliorhinus canicula from the Central Tyrrhenian Sea"

_antioxidants, 2023, doi:10.3390/antiox12020524_

Round 1

Reviewer 1 Report

The main problem is with specification samples of muscles. It should be better specified examined samples of muscles, for example like in: Bakhshalizadeh, S.; Mora-Medina, R.; Fazio, F.; Parrino, V.; Ayala-Soldado, N. Determination of the Heavy Metal Bioaccumulation Patterns in Muscles of Two Species of Mullets from the Southern Caspian Sea. Animals 2022, 12, 2819. - it was shown it this study differences between metal concentration in dependence on localisation of the prepared muscles: ventral, dorsal or caudal. The another possibility, that the authors collected the whole skeletal muscles of the sharks, but it should be written. The same problem seems to be in measuring oxidative stress evidences – the samples of muscles should be better specified.

In the study, the authors compared cumulation of the metals with other animals from Mediterranean Sea (lines 271-277) but it should be depicted some important information, about differences in measurements. For example, in the case of the  Galeus melastomus (study of Gaion et al., 2016) is depicted that samples of muscles were 5g portions taken from the area near dorsal fin, and the animals were collected in April. The samples were dried in mixture of HNO3 and H2O2 and the metal concentration were measure with ASS technique.  So, some differences in collection, digestion and measurement of the samples occurred when compared with methods presented  in this manuscript. Especially that the authors underlined that “our results can be considered as a reference point for future investigations on benthic or demersal small-sized mesopredator sharks coming from other Mediterranean areas, and naturally exposed to aquatic contaminants.”

Author Response

The main problem is with specification samples of muscles. It should be better specified examined samples of muscles, for example like inBakhshalizadeh, S.; Mora-Medina, R.; Fazio, F.; Parrino, V.; Ayala-Soldado, N. Determination of the Heavy Metal Bioaccumulation Patterns in Muscles of Two Species of Mullets from the Southern Caspian Sea. Animals 2022, 12, 2819. - it was shown it this study differences between metal concentration in dependence on localisation of the prepared muscles: ventral, dorsal or caudal. The another possibility, that the authors collected the whole skeletal muscles of the sharks, but it should be written. The same problem seems to be in measuring oxidative stress evidences – the samples of muscles should be better specified.

In the study, the authors compared cumulation of the metals with other animals from Mediterranean Sea (lines 271-277) but it should be depicted some important information, about differences in measurements. For example, in the case of the  Galeus melastomus (study of Gaion et al., 2016) is depicted that samples of muscles were 5g portions taken from the area near dorsal fin, and the animals were collected in April. The samples were dried in mixture of HNO3 and H2O2 and the metal concentration were measure with ASS technique.  So, some differences in collection, digestion and measurement of the samples occurred when compared with methods presented  in this manuscript. Especially that the authors underlined that “our results can be considered as a reference point for future investigations on benthic or demersal small-sized mesopredator sharks coming from other Mediterranean areas, and naturally exposed to aquatic contaminants.”

Re. We thank the reviewer for his/her observation. The analyses, both in term of metal accumulation and oxidative status, were performed on samples of muscle taken from the area near the dorsal fin. We added this information in the Material and Methods section.

We agree with the referee that differences in collection, digestion and measurement of the samples may influence the results. Thus, as stated above, we provided information on samples collection, and changed the sentence: “Our results can be considered as a reference point for future investigations on benthic or demersal small-sized mesopredator sharks coming from other Mediterranean areas, and naturally exposed to aquatic contaminants” with “Our results may be considered a reference point for future investigations through ICP-MS technique on small benthic or demersal mesopredator sharks coming from other Mediterranean areas, and naturally exposed to aquatic contaminants”.

Reviewer 2 Report

The article presents important environmental and toxicological studies. It should be published, and authors should consider of the following notes.

L53 - explain abbreviations,

L67 - explain the abbreviation GPx

L292 - similar relationships were also noted for freshwater fish, e.g. The European Zoological Journal, 88:1; 1084-1095 DOI: 10.1080/24750263.2021.1988160 (please add)

L 383 - should be ... energy release.

Author Response

The article presents important environmental and toxicological studies. It should be published, and authors should consider of the following notes.

L53 - explain abbreviations

Re. Done

L67 - explain the abbreviation GPx

Re. Done

L292 - similar relationships were also noted for freshwater fish, e.g. The European Zoological Journal, 88:1; 1084-1095 DOI: 10.1080/24750263.2021.1988160 (please add)

Re. Thank you for the suggestion. However, we preferred to use the reference in the more general context of the introduction.

L 383 - should be ... energy release.

Re. Done

Reviewer 3 Report

The article, Contamination by Trace Elements and Oxidative Stress in the Skeletal Muscle of Scyliorhinus canicula from the Central Tyrrhenian Sea by Mariacristina et al. is well written and provides interesting new information which has bearing on important aspects of seafood safety and risk assessments.

The experimental design is sound, the findings well described, and the results support the conclusions. Only minor modifications are required for publication. The content described in the major comments is for the authors to consider applying in their future work.

Major comments:

Since selenoenzymes such as glutathione peroxidase (GPx) require adequate amounts of biologically available selenium, but several of the elements studied in the current work are soft electrophiles that bind selenium, it would be valuable for the authors to add selenium to the panel of metals included in future studies. Several selenoenzymes are involved in protection against oxidative damage, but numerous other essential functions are performed by other enzymes in the selenoproteome.

While most famously bound by mercury, an electrophile that strongly reacts with selenium, the most potent intracellular nucleophile, the coaccumulation of selenium in highly stable inorganic forms bound to the indicated ‘heavy” metals is expected to be an important aspect of their toxicity.

Since the ocean fish that marine predators consume will be selenium-rich, those that eat marine mammals can be exposed to heavy metals in molar excess of selenium (potentially exacerbating adverse effects of those exposures), the effects of exposures to soft electrophiles is best evaluated in terms of concomitant selenium intakes. The molar relationships between selenium and mercury as well as other electrophiles in invertebrates remains inadequately studied, but appear likely to provide additional insights.

There are cooperative effects of coexposures to selenium-binding electrophiles that are likely to exacerbate loss of selenoenzyme activities as well as potentially synergistic relationships between those that induce Fenton reactions and those that compromise selenium availability.

The Health Benefit Value (HBV) of various ocean fish is currently established in terms of relative molar quantities of mercury and selenium, but the intended purpose of this index involves inclusion of the full range of electrophilic metallic (and perhaps certain organic) electrophiles. The current work is excellent and has the potential to initiate the expansion of the HBV in this more comprehensive direction.

If you choose to more closely examine effects of size and weight on the parameters being measured, selectively sampling to include sufficient numbers of small and more large fish to obtain a statistically reliable representation of the full dynamic range of the population being studied may prove helpful in future studies.  

Minor Comments:

Unless there is a specific reason which I overlooked, Table 1, 5, and 6 could simply indicate the element symbols rather than specifying the mass of the elemental measured.

Line 195; Reword to: “with males being larger and heavier than females.”

Line 208: Reword middle section of sentence to: “but were significantly lower in specimens” …

Line 223: Replace “was” with “were”.

Line 224: The comma in the value of WBC should perhaps be a period (186.35 instead of 186,35).

Lines 226-230. Usual expressions of p values indicate values less than indicated ranges. It seldom happens that all values would be exactly .0100, so unless these values were uniformly = 0.01, perhaps consider inserting a less than or equal to symbol? If some would be in slight excess, then >0.05 if needed.

Author Response

The article, Contamination by Trace Elements and Oxidative Stress in the Skeletal Muscle of Scyliorhinus canicula from the Central Tyrrhenian Sea by Mariacristina et al. is well written and provides interesting new information which has bearing on important aspects of seafood safety and risk assessments. The experimental design is sound, the findings well described, and the results support the conclusions. Only minor modifications are required for publication. The content described in the major comments is for the authors to consider applying in their future work.

Major comments:

Since selenoenzymes such as glutathione peroxidase (GPx) require adequate amounts of biologically available selenium, but several of the elements studied in the current work are soft electrophiles that bind selenium, it would be valuable for the authors to add selenium to the panel of metals included in future studies. Several selenoenzymes are involved in protection against oxidative damage, but numerous other essential functions are performed by other enzymes in the selenoproteome. While most famously bound by mercury, an electrophile that strongly reacts with selenium, the most potent intracellular nucleophile, the coaccumulation of selenium in highly stable inorganic forms bound to the indicated ‘heavy” metals is expected to be an important aspect of their toxicity. Since the ocean fish that marine predators consume will be selenium-rich, those that eat marine mammals can be exposed to heavy metals in molar excess of selenium (potentially exacerbating adverse effects of those exposures), the effects of exposures to soft electrophiles is best evaluated in terms of concomitant selenium intakes. The molar relationships between selenium and mercury as well as other electrophiles in invertebrates remains inadequately studied, but appear likely to provide additional insights.

There are cooperative effects of coexposures to selenium-binding electrophiles that are likely to exacerbate loss of selenoenzyme activities as well as potentially synergistic relationships between those that induce Fenton reactions and those that compromise selenium availability.

The Health Benefit Value (HBV) of various ocean fish is currently established in terms of relative molar quantities of mercury and selenium, but the intended purpose of this index involves inclusion of the full range of electrophilic metallic (and perhaps certain organic) electrophiles. The current work is excellent and has the potential to initiate the expansion of the HBV in this more comprehensive direction.

If you choose to more closely examine effects of size and weight on the parameters being measured, selectively sampling to include sufficient numbers of small and more large fish to obtain a statistically reliable representation of the full dynamic range of the population being studied may prove helpful in future studies.

Re. We tank the reviewer for her/his suggestions. We will keep them in mind in our future studies.

Minor Comments:

Unless there is a specific reason which I overlooked, Table 1, 5, and 6 could simply indicate the element symbols rather than specifying the mass of the elemental measured.

Re. Done

Line 195; Reword to: “with males being larger and heavier than females.”

Re. Done

Line 208: Reword middle section of sentence to: “but were significantly lower in specimens” …

Re. Done

Line 223: Replace “was” with “were”.

Re. Done

Line 224: The comma in the value of WBC should perhaps be a period (186.35 instead of 186,35).

Re. Done

Lines 226-230. Usual expressions of p values indicate values less than indicated ranges. It seldom happens that all values would be exactly .0100, so unless these values were uniformly = 0.01, perhaps consider inserting a less than or equal to symbol? If some would be in slight excess, then >0.05 if needed.

Re. Done

Reviewer 4 Report

Dear authors,

Congratulations for your work!

I have some recommendations before your manuscript is accepted for publication:

1. replace the word sex/sexes with gender/genders

2. figure 1 need to give more details about the geographic locations of the sampling area. Could you add the coordinates and the name of the surroundings, at least?

3. As reader, I was interested in see a table summarizing the data to characterize the 24 specimens, as mentioned in paragraph 108-115, as follows:

Sample no

Gender

Weight

Length

Maturity

1

...

24

4. The results from table 1 and 2 would be better represented as graphs. did you try to represent these results as graphs, why did you choose to present them as table? Is there any particular reason for your choice?

5. I think there are too many references (99) from which 25 are from the last 5 years. Please adjust this section and pay attention to the instructions how to write the references.

6. In the Abstract and Conclusions sections I would like to see some figures: how much higher is the measured concentration compared to those from liver or skin.

7. The paper would be more interesting if the sources of pollution with heavy metals would be described and associated to your research. Could you add a paragraph in the Discussion section about the sources of pollution?

Good luck!

Author Response

Dear authors,

Congratulations for your work!

I have some recommendations before your manuscript is accepted for publication:

  1. replace the word sex/sexes with gender/genders

Re. We prefer to maintain the terms sex/sexes because of their biological relevance.

  1. figure 1 need to give more details about the geographic locations of the sampling area. Could you add the coordinates and the name of the surroundings, at least?

Re. As required, the geographic coordinates of the sampling area have been included in the text (see 2.1 Sampling collection).

  1. As reader, I was interested in see a table summarizing the data to characterize the 24 specimens, as mentioned in paragraph 108-115, as follows: Sample no Gender Weight Length Maturity

Re. A table which summarizes the data related to the 24 specimens enrolled in the study has been now provided in supplementary materials.

  1. The results from table 1 and 2 would be better represented as graphs. did you try to represent these results as graphs, why did you choose to present them as table? Is there any particular reason for your choice?

Re. Table 1 and 2 are only intended to represent the values of trace elements concentrations (Table 1) and of oxidative stress indicators (Table 2) in the muscle tissue of the lesser spotted dogfish S. canicula, and not to provide a direct comparison among them, which would be difficult to read in a graph even considering the different measurement units. For these reasons, we would prefer to use this representation which has already been approved by other 3 referees.

  1. I think there are too many references (99) from which 25 are from the last 5 years. Please adjust this section and pay attention to the instructions how to write the references.

Re. As suggested by the referee, several references, not strictly necessary, have been delated.

The reference section has been formatted according to the journal's instructions.

  1. In the Abstract and Conclusions sections I would like to see some figures: how much higher is the measured concentration compared to those from liver or skin.

Re. Thanks for the suggestion, but unfortunately, we currently have no figures to include in the Abstract and Conclusions sections.

  1. The paper would be more interesting if the sources of pollution with heavy metals would be described and associated to your research. Could you add a paragraph in the Discussion section about the sources of pollution?

Re. We thank the referee for his/her suggestion. To know the sources of pollution impacting on our study area would be very interesting and would give a higher value to our paper. Unfortunately, information on this issue is not yet available.